# Exploring the Relationship among Human Activities, COVID-19 Morbidity, and At-Risk Areas Using Location-Based Social Media Data: Knowledge about the Early Pandemic Stage in Wuhan

**DOI:** 10.3390/ijerph19116523

**Published:** 2022-05-27

**Authors:** Mengyue Yuan, Tong Liu, Chao Yang

**Affiliations:** 1School of Remote Sensing and Information Engineering, Wuhan University, Wuhan 430072, China; mengyueyuan@whu.edu.cn (M.Y.); tongl@cug.edu.cn (T.L.); 2National Engineering Research Center for Geographic Information System, China University of Geosciences (Wuhan), Wuhan 430079, China

**Keywords:** COVID-19 morbidity, human-activity patterns, social media check-ins, spatial analysis

## Abstract

It is significant to explore the morbidity patterns and at-risk areas of the COVID-19 outbreak in megacities. In this paper, we studied the relationship among human activities, morbidity patterns, and at-risk areas in Wuhan City. First, we excavated the activity patterns from Sina Weibo check-in data during the early COVID-19 pandemic stage (December 2019~January 2020) in Wuhan. We considered human-activity patterns and related demographic information as the COVID-19 influencing determinants, and we used spatial regression models to evaluate the relationships between COVID-19 morbidity and the related factors. Furthermore, we traced Weibo users’ check-in trajectories to characterize the spatial interaction between high-morbidity residential areas and activity venues with POI (point of interest) sites, and we located a series of potential at-risk places in Wuhan. The results provide statistical evidence regarding the utility of human activity and demographic factors for the determination of COVID-19 morbidity patterns in the early pandemic stage in Wuhan. The spatial interaction revealed a general transmission pattern in Wuhan and determined the high-risk areas of COVID-19 transmission. This article explores the human-activity characteristics from social media check-in data and studies how human activities played a role in COVID-19 transmission in Wuhan. From that, we provide new insights for scientific prevention and control of COVID-19.

## 1. Introduction

Since the outbreak in Wuhan in December 2019, COVID-19 has rapidly spread worldwide and has become a significant global health event [1,2]. Human activities play an important role in the transmission of infectious diseases. Previous studies have demonstrated that population mobility and activity patterns are essential determinants of the spread of COVID-19 [3,4,5]. Mu et al. [6] used mobile data to model the spread of COVID-19 in Shenzhen, China, and found that the decline in intra-city mobility had a significant impact on the control of COVID-19 transmission. Chang et al. [7] combined mobile-phone track data from the United States with the SARS-CoV-2 propagation model to identify potential high-risk sites and at-risk populations, and they found that a small number of POIs at risk of super-transmission accounted for a large majority of infections. Understanding the relationship between human activity and the risk of COVID-19 transmission is critical. It can help us understand the impact of risk factors on people’s activities on the spread of diseases in the early pandemic stage and improve people’s risk awareness to reduce dangerous exposure. In addition, the new knowledge generated can provide reference information for the government’s decision making relative to COVID-19 and provide scientific advice for reopening public places or other prevention and control measures in the future. Improving the pertinence of measures would also help alleviate the public’s virus-fatigue emotion [8].

While the impact of human activity on the risk of COVID-19 transmission is critical, the quantitative measures for human geographic activities are limited. With the help of geographic big data, we can have a new perspective on infectious disease epidemiology and public health research [9,10]. Geographic big data increase the accessibility over space and time and expand the possibility of infectious disease surveillance, forecasting, and multiscale coordination [11]. Geospatial data services play a vital role in public health surveillance, such as pandemic isolation tracking [12] and epidemic-trajectory monitoring [13]. The crowdsourcing approach using social media and mobile apps presents new solutions for collecting real-time, large-scale, and accurate human movement data. Geographically located posts on social media services such as Twitter, Facebook, and Sina Weibo help characterize disease distribution and reflect public knowledge, attitudes, and behaviors critical in the early stages of the outbreak [14]. For example, Twitter and Sina Weibo have been used to study public attention [15], epidemic prediction [16,17], and people’s sentiment analysis during the COVID-19 epidemic [18,19].

Previous research studies have revealed deep insights into human behavior in the context of COVID-19; still, the spatial relationship between human activities and the spread of COVID-19 has not been fully explored. This paper introduces the relationship among human activities, COVID-19 morbidity, and at-risk areas in the early COVID-19 pandemic stage in Wuhan. In this study, we try to answer the following questions: (1) Which characteristics of human activities are highly associated with the transmission of COVID-19? (2) How to characterize the COVID-19 spatial spread pattern caused by human activities and identify potential high-risk places? To answer these questions, we performed the following: (1) We characterized groups’ activity patterns in different residential areas using Weibo check-in data during the initial stage of COVID-19 prevalence in Wuhan. (2) Focusing on the activity characteristics and the demographic factors of groups in different residential areas, we conducted a spatial regression analysis to examine how these factors shaped the patterns of COVID-19 morbidity in the early pandemic stage in Wuhan. (3) A fine-grained spatial-interaction network was constructed through social media check-in data, which captured the movement from high-morbidity residential areas to public POIs, and we identified the potential at-risk places of transmission.

This study presents COVID-19 transmission and morbidity from December 2019 to January 2020 in Wuhan. With social media check-in data, we characterize Wuhan citizens’ activity patterns and the relationship between human mobility and the spread of COVID-19 in the early pandemic stage. The scenario in Wuhan provides a vital social model to study the pandemic and related human-activity information within a short time frame. We use social media check-in data to construct a comprehensive analysis framework of influential activity factors with spatial autocorrelation. This article provides statistical evidence for the utility of social media check-in data in COVID-19 transmission analysis, which has potential in the epidemiological modeling of infectious diseases.

## 2. Materials and Methods

### 2.1. Study Area

Wuhan, the capital city of Hubei Province, was the first megacity to suffer from the outbreak of COVID-19. Wuhan City has 13 administrative districts for a total area of 8569 km^2^ and a population of over 11.2 million. The administrative divisions of Wuhan include central districts and suburban districts. As shown in Figure 1, Wuhan’s central districts are densely populated and economically developed areas and include seven districts: Jiang’xia District, Jianghan District, Jiangkou District, Hanyang District, Wuchang District, Qingshan District, and Hongshan District. The population in the central districts accounts for 61.1% of the household population of the whole city population. The suburban district area includes six administration districts: East-West Lake District, Hannan District, Caidian District, Jiangxia District, Huangpi District, and Xinzhou District. The population density in the central urban areas is approximately 6968 people$/{\mathrm{km}}^{2}$, and in the non-central urban areas, it is 2895 people$/{\mathrm{km}}^{2}$. From the outbreak of COVID-19 in Wuhan in December 2019 to 16 April 2020 (no new cases), there were, in total, 50,333 diagnosed COVID-19 patients in Wuhan.

### 2.2. Study Design

Our research process is as Figure 2 shows. Firstly, we collected Sina Weibo users’ residential information from their check-in location and labeled these users as different resident groups. Then, we mined the resident groups’ activity indicators and characteristics from their Weibo check-in posts from December 2019 to January 2020. We adopted four indicators to present human activity: activity type, spatial-interaction frequency, outside-activity duration, and gyration radius of group activity. The demographic factors were considered as the independent variables of COVID-19 transmission.

Then, we analyzed the correlation coefficient between the group activity factors and the morbidity patterns of COVID-19 at the administrative-district level. We used Spearman’s correlation coefficient to examine the correlation between activity and demographic factors, and COVID-19 morbidity. Explanatory variables with significant statistical correlations were selected for the next step of the research study. Then, we used spatial econometric model methods to study the relationship between group activity factors and COVID-19 morbidity. To solve the multicollinearity among various factors, we used principal component analysis to extract the main comprehensive factors. Considering the spatial relationship of COVID-19 morbidity in the study area, we selected the appropriate spatial regression models according to the diagnostics for spatial dependence of least squares estimate regression. The spatial regression model explained the spatial dependency and obtained a more robust estimation of the influences of population activity and demographic factors on COVID-19 morbidity.

Finally, we tracked 178,721 Weibo users’ trajectories and constructed spatial-interaction networks. These networks captured people’s movement from high-morbidity residential areas to public POI sites. We performed a visual analysis of spatial-interaction networks and high-risk transmission areas. Then, we used hot-spot analysis for high-risk COVID-19 hot-spot identification and visualization.

### 2.3. Data Collection and Processing

#### 2.3.1. Weibo Data Collection and Processing

Sina Weibo (Weibo) is one of the largest Chinese social media platforms. Twitter and Weibo are both focused on online news and social networking, and they allow users to publish and share information with multi-media short messages. As Figure 3 shows, similar to geo-tagged tweets, Weibo check-in data are also geo-tagged through location-based service (LBS)-augmented social media applications. A Weibo check-in can be regarded as a Weibo user’s activity records with time and location information.

We collected Weibo check-in data for our research study through Sina Weibo’s open APIs (http://open.weibo.com, (accessed on 1 May 2020)), and the collection and use of these data (for research purposes only) are in compliance with Chinese ethics and data privacy laws. We obtained a dataset of all Wuhan Weibo users who posted Weibo check-ins from December 2019 to January 2020; after that, we performed a data-cleaning process. Weibo raw data are encoded in JSON (JavaScript Object Notation) format. To facilitate statistics and analysis, we used python scripts to extract critical fields, including user ID, post ID, creation time, geographic coordinates (latitude and longitude), venue name, and category. We, then, stored the data in comma-separated value (CSV) format. The check-in data were pre-processed to avoid noise and invalid records were filtered using the following criteria: (1) Each check-in had to possess the following information: user ID, time, and geo-location (longitude and latitude). (2) The geo-location of the check-in was in the Wuhan area. (3) The creation time of the check-in lay between December 2019 and January 2020. (4) Users who checked in less than once were deleted. Finally, 455,076 check-in records were kept, with 178,721 users.

#### 2.3.2. Demographic Data and COVID-19 Statistics Data

The demographic data in this research study come from Wuhan Statistical Yearbook (2018) and Wuhan Health Yearbook (2018), which contain the population density at each street level and the proportion of the elderly population over 60 years old, which greatly affected the spread of COVID-19. Wuhan COVID-19-diagnosed patients’ data were issued by the Wuhan Municipal Health Commission on 16 April 2020. Since March, the increase in COVID-19 infection in Wuhan City has been well controlled, and the reason why we took data from April 16 is based on the consideration of data accuracy. The confirmed data in the early stage of the spread of the epidemic were duplicated or missing, and the Wuhan Municipal Government revised the number of confirmed cases of new coronary pneumonia on 16 April. Therefore, as of 16 April 2020, the epidemic statistics in Wuhan could reflect the COVID-19 infection situation in various regions of Wuhan in the most accurate and timely manner. Although the data at the administrative-district level cover a large area, the data have relatively high accuracy when reflecting the overall distribution of COVID-19 infection in Wuhan. Correspondingly, when mining group activities, a larger-scale spatial boundary easily captures larger sample numbers, and it is also more robust for characterizing group activity patterns. Therefore, we used district-level statistics of COVID-19 cases to evaluate the association with group activity patterns. For the risk of COVID-19 infection in each region, we used the morbidity rate for its evaluation.

We also obtained the confirmed COVID-19 cases’ data at the residential-community level from the Center for Disease Control and Prevention of Wuhan. In our research study, these data were used to investigate residential areas with high COVID-19 risk at a fine-grained spatial level, preparing for the subsequent study of the spatial interaction between residential areas with high epidemics and specific POI sites. We divided the region of Wuhan using a 1 km × 1 km regular spatial grid. Based on the number of confirmed COVID-19 cases in each residential community, we calculated the cumulative number of confirmed cases in each area grid and obtained the residence area in Wuhan with high COVID-19 incidence.

#### 2.3.3. Characterizing Human-Activity Factors from Weibo Check-Ins

We used Weibo check-in data to characterize Weibo user groups and their activity characteristics. First, we inferred Weibo users’ residential information based on the POI information containing residential-area information in the Weibo check-in data. We collected 30,276 Weibo users who posted Weibo information in 1644 residential POIs and mapped them to the administrative-district scale. It should be noted that residential information was detailed at the street and community level and did not involve private information such as a user’s detailed home address.

Based on the spatio-temporal and POI information in Weibo check-ins, we characterized the activity factors of groups in different residential areas:Determine the user’s outside-activity type according to the POI category attribute of Weibo check-ins;Calculate the spatial-interaction frequency between residential space and activity space based on the geolocation of Weibo check-ins;Infer the user’s outside-activity duration based on Bayesian inference and the Markov Chain Monte Carlo (MCMC) method;Calculate the radius of gyration of the user’s movement based on the geolocation of Weibo check-ins.

##### Activity Types

Understanding the impact of place on disease transmission is a crucial element of epidemiological research [20]. We can understand a user’s activity venue type from the check-in POIs of social media check-in data. Some users may choose to check-in at specific POIs. Weibo provides a detailed POI classification system for POIs to distinguish the category attributes of each POI. We focused on the POIs of catering services, shopping services, recreation services, and transportation facilities services, which are indoor places that are prone to crowd contact. The intensity of such activities was characterized by calculating the activity type’s proportion in all check-in activities.

In Figure 4, we present the distribution of different kinds of activities in Wuhan before and after the non-pharmaceutical interventions, respectively. After the non-pharmaceutical interventions on COVID-19 and the lockdown policies of Wuhan, the intensity of those activities was dramatically reduced. It means human activities in the two months before the non-pharmaceutical interventions on COVID-19 may have been mainly related to the spread of the epidemic, so we selected the activity characteristics of the two months of December 2019–January 2020 for analysis.

##### Spatial-Interaction Frequency

Population mobility is crucial for predicting the spatial spread of infectious diseases and the decision making behind control measures [21]. As the link among Wuhan’s different areas, human mobility reflects the interactive relationship among urban spaces. Spatial interaction can be quantified by using the users’ trajectories of social media check-in data to calculate the spatial-interaction frequency between the residential space and the activity space in Wuhan. It is not easy to accurately distinguish whether the social function attribute of city area is residential space or activity space. However, we can approximately measure them using the POI information of Weibo check-in data.

We divided the Wuhan city area using regular space grids of 1 km × 1 km and obtained the distribution of residential quarters within the grid space of each area by spatial overlay analysis, and the area grid containing residential quarters was regarded as a unit of residential space. If a space grid contained the POI mentioned in the activity types in the previous step, it was regarded as an activity space. Some space units may belong to both residential space and activity space.

By tracking the check-in trajectories of Weibo users, we calculated the number of times the residential space unit and the activity space unit appeared in the same user’s check-in trajectories. In order to prevent the influence of the total number of Weibo check-ins of different regions, the statistical interaction between each residential space unit was divided by the total number of check-ins of users in this area. The spatial-interaction frequency of each residential space unit was calculated and then aggregated to the administrative-district level. The higher the spatial-interaction frequency means, the closer the contact among people’s check-in activities in the residential space and social public activity places.

##### Duration of Activities

The duration of activities impacts the probability of the transmission of infectious diseases among locations [22]. Users’ outside-activity duration calculates the average time of users’ check-in activities outside the residential area. Because it is difficult to obtain the duration of the activity directly from the check-in data, we used a Bayesian method, forecasting the transition time between a series of activities in the trajectory, approximating the duration of the activities. Activity duration follows a Weibull distribution, which has been demonstrated in empirical studies [23], with the following equation:(1)$$f\left(x;\lambda ,k\right)=\frac{k}{\lambda }{\left(\frac{x}{\lambda }\right)}^{k-1}e^{-{\left(\frac{x}{\lambda }\right)}^{k}}$$
where *λ* is the scale parameter, and *k* is the shape parameter. To estimate the parameters, we used the Markov Chain Monte Carlo (MCMC) method. MCMC is a common tool in Bayesian statistical calculation. The MCMC algorithm can generate many samples from the given probability distribution and estimate the parameters based on these samples [24]. We extracted the direct conversion time from the Weibo check-in data, which approximates the activity duration, as the data samples to train the distribution in the above formula. Then, we calculated the probability distribution function of the duration of each type of activity. We considered POI types other than residential places (including catering services, shopping services, recreational services, and transport services) as outside check-in activities, weighted by the proportion of the number of various activities to the total number of check-ins, and we calculated the average duration of activities for each regional user group.

##### Radius of Gyration

As a popular mobility measure, the radius of rotation of user movement has been widely applied to represent human flow patterns [25]. The radius of gyration represents the standard deviation of the distance between the points on the trajectory and the center of mass. It reflects the range of users’ activities through the geographical distribution of the check-in locations and their frequency of visits. The ROG values are defined as follows:(2)$$\mathrm{ROG}=\sqrt{\frac{1}{m}\cdot \sum_{i=1}^{m}{\left(p_{i}-p_{c}\right)}^{2}}$$
where m represents the number of check-in records of the user; $p_{i}$ indicates the user’s *i*-th check-in location; and $p_{c}$ is the center point of all check-in locations. To extract this metric, the model calculates the user’s check-in activity’s geographic centroid; then, it obtains the distances between the user’s check-in points and the centroid and calculates the radius of gyration of the user group in the dataset using Equation (2).

#### 2.3.4. Factors Associated with COVID-19 Morbidity Rate

As Table 1 shows, we listed possible human activity and demographic factors associated with the COVID-19 morbidity rate. As shown in the upper part of Section 2.3.2, we extracted the key features of user activities from social media check-in data, including the type of user activities, the spatial-interaction frequency, the duration of outside activities, and the radius of gyration of movement. We explain the reasons for the selection of activity indicators and the calculation method in Section 2.3.2. The driving force behind the spread of the COVID-19 pandemic is complex. While we are concerned about human activity, the active groups’ demographic factors also need to be considered, as demographics are essential factors influencing infectious diseases [26]. The resident-population density and aging degree were considered as the demographic characteristics for the administrative districts in our study.

### 2.4. Spearman’s Correlation

Spearman’s correlation coefficient is widely used to evaluate the correlation analysis between the severity of epidemics and related factors [27,28]. There may be a non-Gaussian normal distribution, a spatial autocorrelation, and a possible non-linear relationship between the COVID-19 morbidity data and Weibo check-in activity factors. Therefore, this study used Spearman’s correlation coefficient to make a preliminary correlation assessment of each factor. This step’s aim is to select relevant variables for the subsequent spatial regression analysis. The defining equation of Spearman’s rank correlation coefficient is as follows:(3)$$\rho =\frac{\sum_{i=1}^{n}\left(x_{i}-\bar{x}\right)\left(y_{i}-\bar{y}\right)}{\sqrt{\sum_{i=1}^{n}{\left(x_{i}-\bar{x}\right)}^{2}}\sqrt{\sum_{i=1}^{n}{\left(y_{i}-\bar{y}\right)}^{2}}}=\frac{cov\left(x,y\right)}{S_{x}S_{y}}\;,$$
where *n* is the total number of data samples;$\;x_{i}$ and $y_{i}$ are the ranks of variables *x* and *y*; $\;\bar{x}$ and $\bar{y}$ are the average ranks of *x* and *y*;  cov(x,y) is the variance of *x* and *y*; $\;S_{x}$ and $S_{y}$ are the products of the standard deviation of *x* and *y*; and ρ represents Spearman’s rank correlation coefficient. The value of this coefficient ranges from −1 to 1. The larger the absolute value of Spearman’s rank correlation coefficient, the stronger the correlation between the two variables.

### 2.5. Spatial Regression Models

Spatial data usually present a certain degree of positive spatial autocorrelation. In the study of spatial epidemiology, the distribution and transmission of infectious diseases are usually spatial processes [29]. Due to the possible spatial effect of infectious diseases, we used spatial regression models to evaluate how the demographic and activity factors shaped the patterns of COVID-19 morbidity in the early pandemic stage in Wuhan.

The spatial lag model (SLM) considers the spatial spillover effect of the dependent variable and adds the dependent variable’s spatial lag term to the classic linear regression model^30^. The model is expressed as:(4)$$y_{i}=\beta_{0}+x_{i}\beta +\rho W_{i}y_{i}+\varepsilon_{i}$$
where i denotes an administrative district; $y_{i}$ indicates the COVID-19 morbidity rate of the *i*-th district; $x_{i}$ indicates the selected explanatory variable; β represents the regression coefficient; $\beta_{0}$ represents the intercept; ρ is the spatial lag parameter; *W* is the *n* × *n* spatial weight matrix; $W_{i}$ represents the vector in the spatial weight matrix; and $\varepsilon_{i}$ is a random error term. In our research study, the SLM was used as a spatial model to examine how the COVID-19 infection situation in an area was affected by the neighboring area.

The spatial error model (SEM) reflects the spatial dependence effect through the spatial autocorrelation setting of the error terms, when the relationship between adjacent units and this unit may also be expressed by some unobserved or omitted variables [30]. The model assumes spatial dependence in the error term, which is defined as:(5)$$y_{i}=\beta_{0}+x_{i}\beta +\lambda W_{i}\xi_{i}+\varepsilon_{i}$$
where λ is the spatial error coefficient that represents the spatial dependence of residuals, and $\xi_{i}$ is the component of the error term. The remaining symbols have the same meaning as those in Formula (4). The spatial error model estimates the correlation between the residuals of each region and the residuals of adjacent regions.

In order to avoid the estimation deviation caused by the spatial correlation, we tested the spatial correlation of COVID-19 morbidity using global Moran’s I. For selecting spatial model types, we referred to the selection criteria summarized by Anselin [30] and selected the appropriate model based on the results of the Lagrange multiplier test.

In the process of multiple linear regression, the problem of multicollinearity is prone to appear. The correlation between factors makes model estimation distorted or difficult to be accurately performed [31]. The correlation between explanatory variables can be tested by the correlation coefficient matrix of independent variables, the Kaiser–Meyer–Olkin (KMO) test, and Bartlett’s spherical test. The independent variables we selected involved activity and population attributes, and there was, inevitably, multicollinearity between variables. To address the issue of multicollinearity between independent variables and to comprehensively analyze the various factors that influenced the spread of COVID-19, we employed principal component regression (PCR). This analytical approach combines principal component analysis (PCA) with multiple linear regression. PCA can transform the original variables into several influential factors that are uncorrelated with each other; we can, then, eliminate multicollinearity in multiple linear regression [32]^.^ This method retains most of the information on the primary factors and makes the feature dimensions irrelevant to each other.

In this study, in order to avoid the adverse effects brought by the correlation of explanatory variables, we used PCA to standardize the explanatory variables, extracted the unrelated principal components, and obtained the factor-score function of each principal component with respect to the original explanatory variables. The integrated principal component variables were used to replace the original variables and to retain the main information of the original variables. A regression analysis was, then, performed with the extracted principal components as independent variables and the COVID-19 morbidity rate as the dependent variable to obtain the regression model. We compared the spatial regression models and chose the best fitting model. Lastly, we substituted the factor-score function into the model to obtain the quantitative stable regression relationship of the COVID-19 morbidity rate with the original independent variables after standardization.

### 2.6. Spatial-Interaction Matrix Modeling

After discussing the activity characteristics related to the COVID-19 regional morbidity rate, we performed spatial-interaction modeling to analyze the COVID-19 transmission patterns in more detail. We tracked the check-in trajectory data of Weibo users during the COVID-19 large-scale transmission period (from December 2019 to January 2020), analyzed the spatial spread of COVID-19 within the city, and identified potential places with high transmission risk. This section focuses on the spatial interaction between residential areas with high epidemics and specific public POI sites. We tried to locate high-risk places for virus transmission using the intensity of the interaction between residential areas and specific activity venues.

To explore Wuhan’s spatial propagation process at a finer granularity, we divided Wuhan’s region using 1 km × 1 km regular spatial grids. We used the number and location information of the COVID-19 confirmed cases notified by each residential community to calculate the cumulative number of confirmed cases in each area grid. Area grids where the cumulative number of confirmed cases exceeded 3 were considered high-incidence residential areas. Based on our previous analysis of activity factors’ influence, we mainly focused on the types of activities closely related to COVID-19 (including catering, shopping, traffic).

We designed an algorithm to construct an internal-interaction matrix formed by check-in POI sites and high-incidence residential areas, and we evaluated the intensity of interaction between specific urban spaces based on user trajectories. Suppose the collection of residential-space units with high COVID-19 incidence is$\;L=\left\{l_{1},l_{2},l_{3},\ldots ,l_{m}\right\}$. The collection of public POI space units is $p=\left\{p_{1},p_{2},p_{3},\ldots ,p_{n}\right\}$. Interaction matrix *I* between the residential space and the POI space is:(6)$$I=(\begin{array}{ccccc} i_{l_{1},p_{1}} & i_{l_{1},p_{2}} & \cdots & i_{l_{1},p_{n-1}} & i_{l_{1},p_{n}}\\ i_{l_{2},p_{1}} & i_{l_{2},p_{2}} & \cdots & i_{l_{2},p_{n-1}} & i_{l_{2},p_{n}}\\ \vdots & \vdots & & \vdots & \vdots \\ i_{l_{m-1},p_{1}} & i_{l_{m-1},p_{2}} & \cdots & i_{l_{m-1},p_{n-1}} & i_{l_{m-1},p_{n}}\\ i_{l_{m},p_{1}} & i_{l_{m},p_{2}} & \cdots & i_{l_{m},p_{n-1}} & i_{l_{m},p_{n}} \end{array})$$
where $i_{l_{m},p_{n}}$ represents the interaction strength between residential space $l_{m}$ and POI space $p_{n}$. After dividing the study area into grids, the original trajectory connection between points is converted into the trajectory connection between grids. The intensity of interaction between grids is determined by the number of times each grid appears on the same user track simultaneously. The more trajectories connect two regions, the closer the interaction between the two regions, and the higher the interaction intensity. We collected the user check-in trajectories within two months, which included user ID, check-in time, check-in geographical coordinates, and POI type. The calculation process of spatial-interaction intensity is as follows:Screen out check-in points located in residential areas with high incidence of COVID-19 and add their corresponding grid numbers to set *L*;Screen out check-in points with POI types belonging to three specific categories and add their corresponding grid numbers to set *p*;Calculate the interaction strength between set-*p* elements and set-*L* elements, that is, the number of simultaneous occurrences in the same track; after traversing all the check-in trajectories of users in Wuhan within two months, spatial-interaction matrix *I* is finally obtained;The cumulative interaction intensity with set *L* is calculated for each grid in set *p*, and the sum of the elements in column *i* of the interaction matrix is the overall interaction intensity between *Pi* and the high-incidence residential areas;Identify the grids that contain POIs with high-risk transmission of COVID-19 based on the overall interaction strength with high-incidence areas.

After constructing the spatial-interaction matrix and identifying the high-risk POI spatial grids, the spatial distribution of the POIs with high-risk transmission was further analyzed. Hot-spot analysis is a method for the identification of statistically significant hot spots. Based on Getis-Ord Gi* statistics, hot-spot analysis identifies statistically significant spatial clusters of high values as hot spots and those of low values as cold spots [33]. We used the Getis-Ord Gi* hot-spot analysis method in GeoDa to analyze the hot-spot spatial distribution of the places with a high transmission risk of COVID-19.

## 3. Results

### 3.1. Assessing the COVID-19 Risk Areas in Wuhan

As Figure 5 shows, the spatial distribution results of the COVID-19 morbidity rate in Wuhan are diverse in the 13 administration areas. The COVID-19 infection rate in Wuhan’s central city was more serious than that in the suburban areas, especially the Hankou area (Jiang’an District, Jianghan District, Qiaokou District) and Hannan District north of the Yangtze river.

We used global Moran’s *I* to evaluate the spatial autocorrelation of the distribution of COVID-19 morbidity rates. The spatial relationship was defined according to the districts’ adjacency relationship with Wuhan, and the queen adjacency weight was selected to calculate the spatial weight matrix; then, global Moran’s *I* was calculated to be 0.543. Moreover, we chose a randomized test process; a total of 999 permutations were used to construct the reference distribution, and the *p*-value was 0.001. This result shows that there was spatial autocorrelation in the distribution of the COVID-19 morbidity rate in Wuhan.

### 3.2. Variable Selection with Spearman’s Correlation

After we listed the factors associated with COVID-19 morbidity, we used Spearman’s correlation coefficient to examine the correlation between the influencing factors and COVID-19 morbidity in different administrative divisions. The results are shown in Table 2. The demographic factors, including PD and AOP, were positively correlated, indicating that higher population density and population ageing may increase the prevalence of COVID-19. Among the activity factors, the morbidity of COVID-19 was positively correlated with POC, POS, POT, SIF, and DOA, which suggests that more frequent catering, shopping, transportation activities, and spatial interaction might have aggravated transmission, and a longer duration of outside activities meant a greater risk of infection. However, POR and ROG had no significant effects on the prevalence of COVID-19. For POR, we found that the intensity of recreational activities in each region was relatively average, so the regional difference in this type of activity was not significant. For ROG, the impact of the range of movement was mainly found to be related to the expansion of the epidemic’s spread, but the range of movement was not the main factor in the increase in the morbidity of COVID-19.

### 3.3. The Relationship between Human Activities Factors and COVID-19 Morbidity Patterns

#### 3.3.1. Principal Component Extraction

Before establishing the principal component analysis model, we tested the correlation among the explanatory variables to illustrate the rationality of the principal component analysis method. The KMO test and Bartlett’s ball test are statistical test methods used to determine whether the original variable is suitable for principal component analysis. Table 3 shows that the value of the KMO test was 0.810, which means that the sum of squares of the simple correlation coefficients was far greater than that of the partial correlation coefficients, and the correlation between variables was strong. Bartlett’s test’s significance was 0.000, which means that the correlation coefficient matrix was not a unit matrix, and there was a correlation between the original variables. These calculation indexes indicate that the data were suitable for principal component analysis.

Then, we conducted a principal component analysis on the PD, AOP, POC, POS, POT, SIF, and DOA factors and transformed them into seven unrelated principal components. Table 4 shows the explained variance and the cumulative contribution rate of each principal component. To explain about 90% of the total variance, we retained the first three principal components (PCs). The first three principal components can explain as much as 93.76% of the information. The component-score coefficient matrix of the first three principal components is shown in Table 5, which reflects the relative importance of each original variable to the three principal components.

#### 3.3.2. Spatial Regressive Model Estimation

As can be seen in Section 3.1, we found a spatial correlation between the COVID-19 morbidity in the districts of Wuhan. We considered the spatial regression model to avoid the estimation error caused by spatial dependence. The Lagrange multiplier statistics are shown in Table 6. Since the Lagrange multiplier (lag) was significant, and the Lagrange multiplier (error) was not significant, it was more appropriate to use the SLM than the SEM in this study.

Table 6 shows the statistical results of the SLM. For comparison, the classic ordinary least squares (OLS) model was also calibrated. We analyzed the fitting degree of the two models; compared with the OLS model, the R-squared of the spatial lag model was significantly improved, indicating that its regression residual was effectively reduced. The absolute value of the log-likelihood function and the values of the Akaike info criterion and Schwarz criterion of the spatial lag model were smaller than those of the OLS, indicating that the performance of the SLM was better than that of the OLS. Therefore, we believe that the spatial lag model has better explanatory power than the OLS linear model. We also performed the Breusch–Pagan test on the SLM to prove that the model’s random error terms satisfy the same variance. The regression results of the SLM showed that the distribution of the prevalence of COVID-19 showed a certain spatial lag effect, and the first three principal components showed a significant positive correlation. We compared the SLM model’s predicted value with the actual distribution of COVID-19 in Wuhan, as shown in Figure 6. The results show that the SLM model can well demonstrate the overall impact among human activity, demographic factors, and the morbidity rate of COVID-19.

According to the component-score coefficient matrix in Table 5, the regression coefficients of the principal components were substituted into the original variables to obtain a new model, and the regression coefficients of the original variables are shown in Table 7. Table 7 shows the quantitative relationship between the influence factors and the morbidity of COVID-19 in the period of COVID-19 large-scale transmission in Wuhan. PD and AOP showed positive correlations, indicating that high population density and population ageing exacerbated regional COVID-19 prevalence. The activity characteristics, including SIF, POC, POS, POT, and DOA, had positive associations with the COVID-19 morbidity rate, reflecting that spatial interaction, types of activities, and the duration of outside activities were significant explanatory variables for the spread of COVID-19. Among the types of activities, the proportion of catering activities had the most significant impact, suggesting that places with small indoor spaces and long face-to-face time were more likely to increase the risk of COVID-19 infection. The above results reflect those demographics and human-activity characteristics that significantly impacted the transmission of COVID-19 in Wuhan.

### 3.4. Spatial Interaction and Uncovering of At-Risk Areas

In Figure 7a, we present the intensity of the spatial interactions between high-morbidity residential areas and public POI sites. By calculating the overall interaction strength between specific public POIs and high-morbidity areas, we mapped the risk of COVID-19 transmission to specific activity spaces, thereby identifying potential POIs with high-risk transmission, as shown in Figure 7b. The spatial interactions between public POI sites and high-incidence residential area show a gathering pattern in the urban center, spreading around the suburbs. Spatial interactions were mainly concentrated within the second ring road of Wuhan. The central urban area is densely populated, and there are a large number of commercial facilities, which correspond to the concentration of highly epidemic residential areas and the gathering area of public POI sites. Therefore, the spatial-interaction intensity of population movement was very high. Spatial interactions spread to the suburbs, reflecting the spatial interactions between the urban center and the suburbs. This mobility pattern reflects the fact that people living in the suburbs often work and socialize in the city’s central area; on the other hand, it reflects that large-scale transportation facilities in the suburbs, such as airports and suburban commercial centers, also attract a number of people. The distribution of high-risk public POI sites is shown in Figure 7b. From 1 December 2019 to 31 January 2020, public POIs with high transmission risk were mainly concentrated in the downtown area along the Yangtze river in Wuhan. The high-risk public POI sites’ spatial distribution was particularly dense in the Hankou area and scattered in the suburbs. The overall distribution was wide, covering multiple centers in the central city area of Wuhan.

To identify statistically significant hot spots, we conducted a hot-spot analysis (Getis-Ord Gi*) on public places at high risk of COVID-19 transmission. The hot-spot analysis results (Figure 8) show that the hot-spot areas (at a 95% confidence level) significant for public places at high risk of COVID-19 transmission were limitedly located in several critical central areas of Wuhan City (areas along the Yangtze river in Jianghan District and Wuchang District, and newly developed Optics Valley) and Tianhe international airport in the northwestern suburbs. Firstly, the areas close to the Yangtze river in Hankou and Wuchang Districts are the center of old city areas in Wuhan. The hot spots significant for high-risk public POI distribution extended from near Hankou Railway Station to the Jianghan Road business district, distributed perpendicularly to the Yangtze river in the northwest–southeast direction and then extended to the central area of Wuchang District on the south bank of the Yangtze river. These areas are commercial POI aggregation centers and highly active places within Wuhan’s second ring. The nearby Huanan seafood market was the site where most of the earliest cases were reported to have been exposed to the virus. Hankou Railway Station is also located near those areas. As an important transportation hub, Hankou Railway Station is also a specific traffic site with high transmission risk and coincides with people frequently coming and going during the Spring Festival transport period. Hankou and Wuchang Districts’ business centers are all areas with large people traffic and convenient transportation in Wuhan. Therefore, they became significant hot areas of high-risk public POI distribution. In addition to the traditional urban center, Optics Valley, a new development area located in the southeast of Wuhan, is also one of the significant hot spots. The rapid development of industry and commerce in this area in recent years has also driven the intensity of population activities. In densely distributed places such as shopping malls, restaurants, and subway stations in the region, population interaction might have increased the risk of COVID-19 transmission. Besides, we also found at-risk places in the city’s outer suburbs, Wuhan Tianhe Airport, an important transportation hub located in the suburbs. Although it is located in the suburbs, the airport is also an indoor space where people gather, especially during the Spring Festival transport period.

## 4. Discussion

Quantitatively analyzing human activity and understanding its relationship with the transmission of COVID-19 are essential to guide the prevention and control of COVID-19. Our study found that the spread of COVID-19 in Wuhan was affected by people’s activities. Specific characteristics of group activities include the intensity of dining, shopping, and transportation; the duration of outside activities; and the spatial-interaction frequency. There are several reasons to explain the correlation between the activity factors we observed and the spread of COVID-19. Catering activities, mainly in places such as restaurants and cafes, are usually performed in an indoor space with a small activity range, and people sharing a meal in a small space can quickly increase the risk of COVID-19 infection. The study by Chang et al. [7] also mentioned that the reopening of restaurants, cafes, and hotels would bring the most significant risk of infection, and reducing the occupancy rate of such venues may significantly reduce the predicted number of infections. Shopping activities are mainly concentrated in commercial areas, which are places with risk of transmission. Especially when the outbreak was approaching the end of the year and the Spring Festival, the frequency of people going to major shopping malls increased. A previous study by Li et al. [34] also showed the impact of the distribution of large-scale shopping and supermarket facilities on the COVID-19 epidemic. The main indoor venues for transportation activities are railway stations, airports, and subway stations, where prevention and control measures of COVID-19 need to be enhanced. As an important transportation hub in central China, Wuhan also coincides with people’s frequent movement during the Spring Festival travel period, making transport locations potential hot spots for transmission. The time people spend at the destination (the duration of travel) also affects the spatial transmission of COVID-19 in urban spaces. In the study by Giles et al. [22], they found that statistical information that ignores travel-time factors may be biased in predicting the spatial spread of infectious diseases.

Regarding the demographic factors, high population density means more people congregate in urban spaces, and high population density accelerates the diffusion of COVID-19 in the population. This is consistent with previous studies by Hirata et al. [35] and Coskun et al. [36].^.^ Population density has been proved to be a significant factor affecting the transmission pattern of COVID-19 before the implementation of social-distancing measures. The infection situation is also related to aging, and the situation of the elderly population may increase the risk of COVID-19 infection. It is because the elderly are susceptible to COVID-19, which is consistent with the current medical research observations [37].

As revealed by the spatial regression model, the spatial lag in the distribution of COVID-19 morbidity reflects the potential spatial spread of infectious diseases. In existing studies, the spatial lag effect of COVID-19 has also been noted [38,39]. The spatial spillover characteristics of COVID-19 mean that the connections between urban areas increased due to the proximity of spatial distance. Areas with high morbidity of COVID-19 have a spatial-spread effect that affects the infection rates in neighboring areas. The spread of epidemics is essentially a spatial process [29], and spatial econometric methods that consider spatial effects could be helpful for research on infectious disease epidemiology.

Based on Weibo users’ check-in trajectories, we showed the impact of human mobility on the spread of COVID-19 from the perspective of spatial interaction and also found some phenomena worth discussing. This study found several high-risk transmission hot spots, including old urban areas and new development areas in Wuhan. Due to the influence of the geographical form of the convergence of the two rivers and the urban development, Wuhan is a typical multi-center city, which may be the internal reason for distributing multiple high-risk transmission hot spots within the city. In addition, there were some spatial interactions between the suburbs and the city center, which means that the confirmed cases living in the suburbs were likely infected in the city’s central area. A similar phenomenon was also observed in the study by Huang et al. [40]. The explanation for this phenomenon is that suburban residents’ activities are characterized by an extensive range of activity space and are highly dependent on the central urban area due to the suburbs’ limited activity venues. This activity characteristic may have expanded the spread of the COVID-19 epidemic, which has been mentioned in a previous study by Zhang et al. [41].

## 5. Conclusions

In this study, we use Sina Weibo data, a widely applied human mobility data source, to explore the relationship between human activity and COVID-19 morbidity patterns and to further understand the mechanisms of impact of human-activity patterns on the spread of COVID-19. The below conclusions were achieved.

(1) The human-activity indicators characterized from Weibo check-in data were shown to have had statistically significant and positive impacts on COVID-19 morbidity in Wuhan. The results provide statistical evidence regarding the utility of human-activity indicators (POC, POS, POT, SIF, and DOA) and demographic factors (PD and AOP) for COVID-19 morbidity patterns in the early pandemic stage in Wuhan.

(2) The COVID-19 morbidity pattern at district level in Wuhan had significant spatial autocorrelation. The SLM explained the spatial dependency and obtained a more robust estimation of the influences of population activity and demographic factors on COVID-19 morbidity.

(3) The spatial-interaction matrix revealed a general transmission pattern within Wuhan and determined the at-risk areas of COVID-19 transmission.

The empirical findings shown in this study can provide helpful insights for policy makers to formulate targeted scientific prevention and control measures in the latest stages of the epidemic. Firstly, human activity is the main factor affecting the spread of COVID-19. The place type and spatial interaction of outside activities were shown to have had considerable influence. For high-risk activity sites, the relevant agencies should strengthen supervision and disinfection in the latest prevention and management measures. Second, the urban center is a high-spread area, and we should pay attention to the strengthening of quarantine measures and crowd control. Besides, people living in the suburbs may have to go to the central city area for some activities, which is a phenomenon worthy of attention. In the long run, the government still needs to take targeted public health measures to minimize the transmission of COVID-19 and the possibility of future outbreaks.

Still, there are several limitations in this study. Similar to other big data, social media check-in data also have some data-quality issues, such as demographic bias and behavioral bias. Most social media users are young people, which means that when we use social media check-in data to study human-activity patterns, elder people groups are neglected. However, data bias is unavoidable in all kinds of user-generated contents [42]. Another problem is the spatio-temporal sparseness of social media check-in data; the personal activity information they provide is incomplete. However, with the support of a large number of data samples, related studies have proved the effectiveness of social media check-in data in analyzing activity behavior and building mobile models [43]. Besides, there may be some deficiencies in the data of COVID-19 confirmed cases in Wuhan. There may be several neglected mild and asymptomatic infections. Some scholars are also worried about the spread of asymptomatic cases [44], but these cases cannot be included in our data. Lastly, we need to illustrate that the transmission-influencing factors during the COVID-19 pandemic were complex and diverse. Our consideration of contributing factors focused on human-activity characteristics and demographics but did not include all real-world characteristics associated with COVID-19 transmission. It should be emphasized that human activity does not fully explain the landscape of COVID-19 transmission, but the model’s prediction accuracy shows that we extensively captured the relationship between human activity and COVID-19 transmission.

## Figures and Tables

**Figure 1 ijerph-19-06523-f001:**
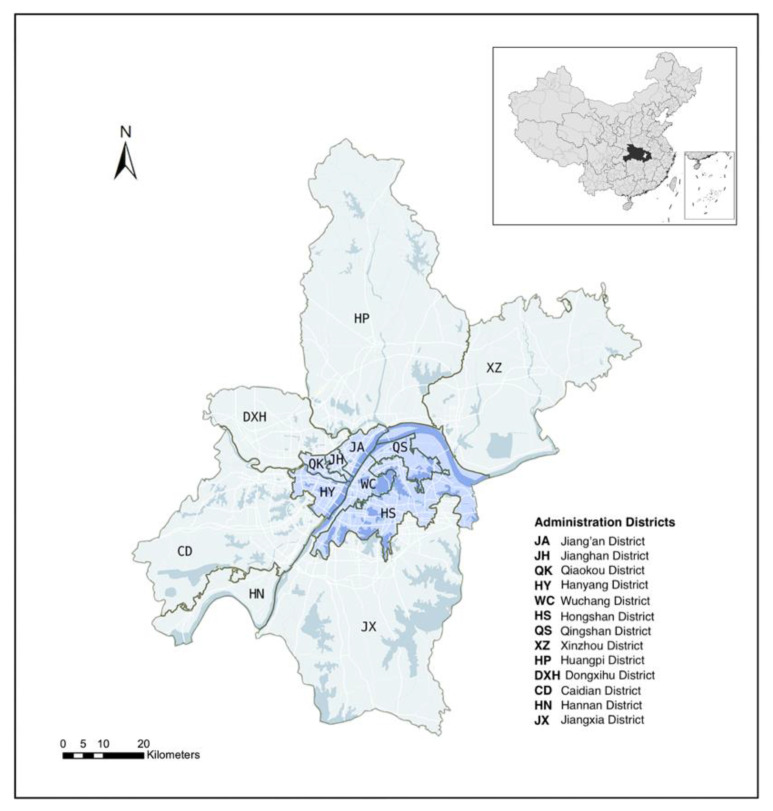
Study area: Wuhan central districts and suburban districts.

**Figure 2 ijerph-19-06523-f002:**
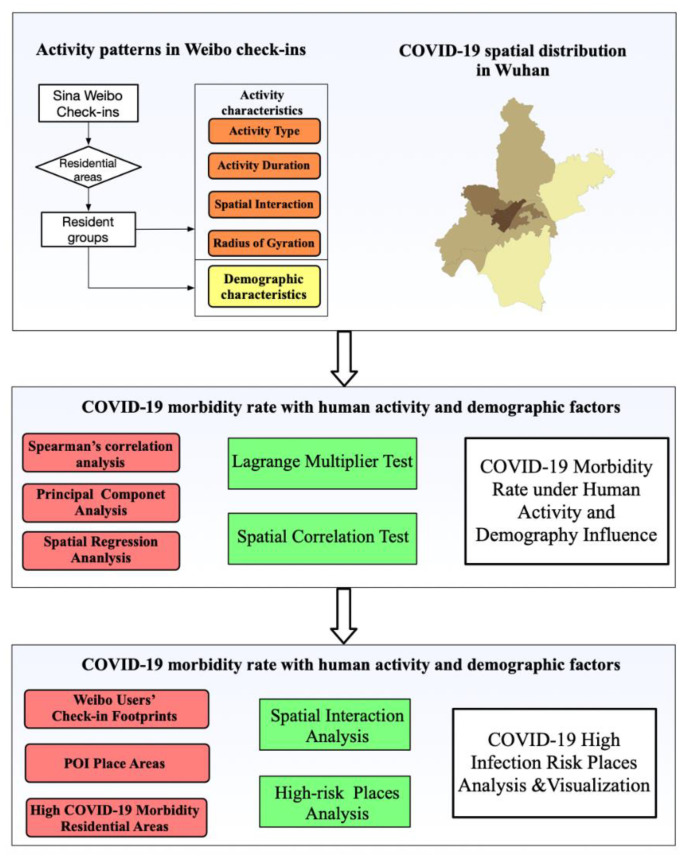
Study design.

**Figure 3 ijerph-19-06523-f003:**
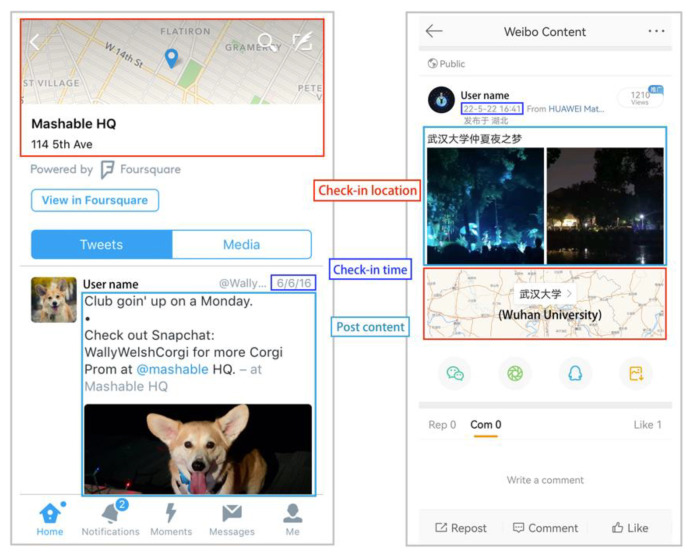
An example of geo-tagged Tweets and Weibo check-ins.

**Figure 4 ijerph-19-06523-f004:**
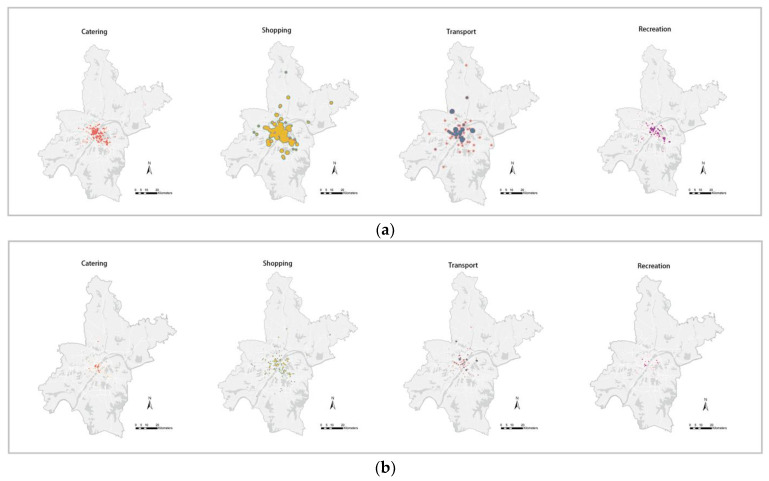
(**a**) Intensity distribution of different activity types (before non-pharmaceutical interventions on COVID-19 in Wuhan). (**b**) Intensity distribution of different activity types (after non-pharmaceutical interventions).

**Figure 5 ijerph-19-06523-f005:**
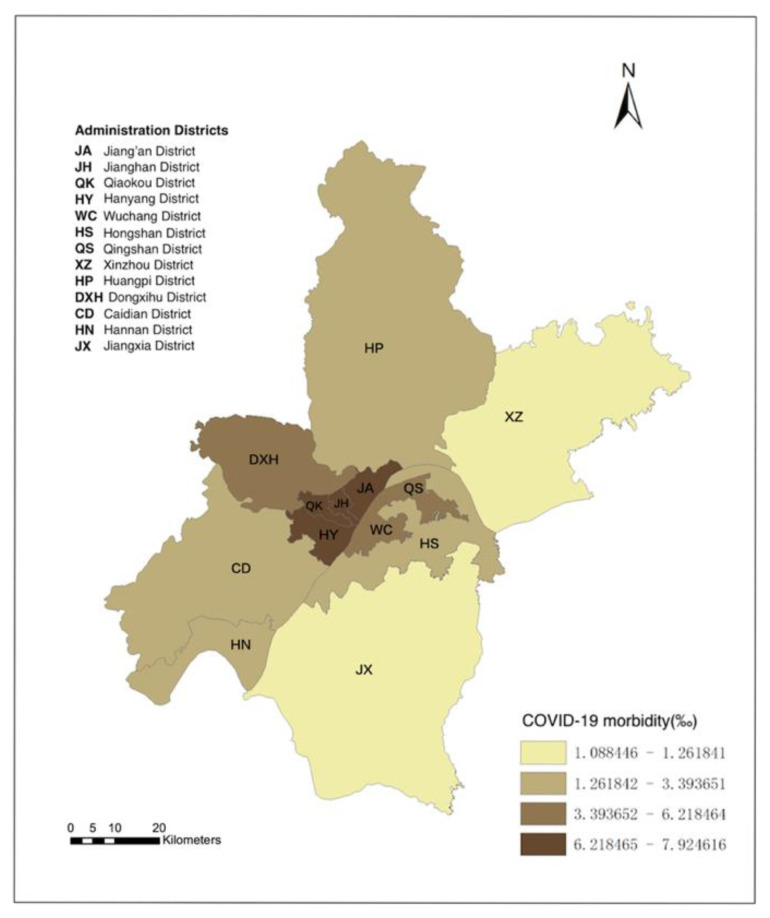
Spatial distribution of COVID-19 morbidity rate in Wuhan.

**Figure 6 ijerph-19-06523-f006:**
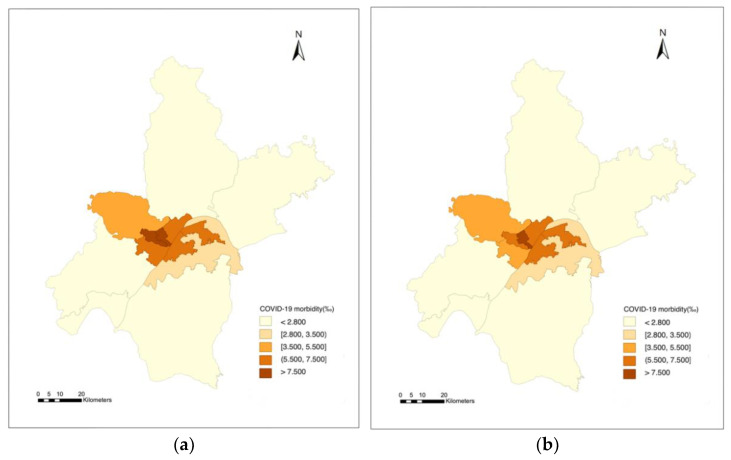
(**a**) Spatial distribution of COVID-19 morbidity rate in Wuhan. (**b**) Spatial distribution of SLM predicted value.

**Figure 7 ijerph-19-06523-f007:**
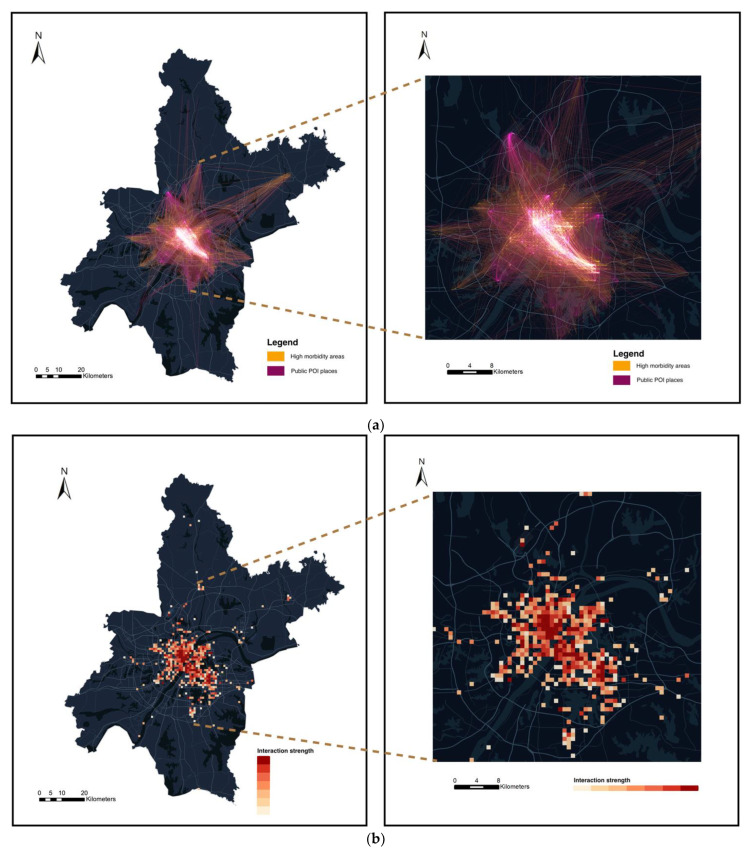
(**a**) Spatial interaction of high-morbidity residential areas and public POI sites. (**b**) Spatial distribution of high-risk public POI sites.

**Figure 8 ijerph-19-06523-f008:**
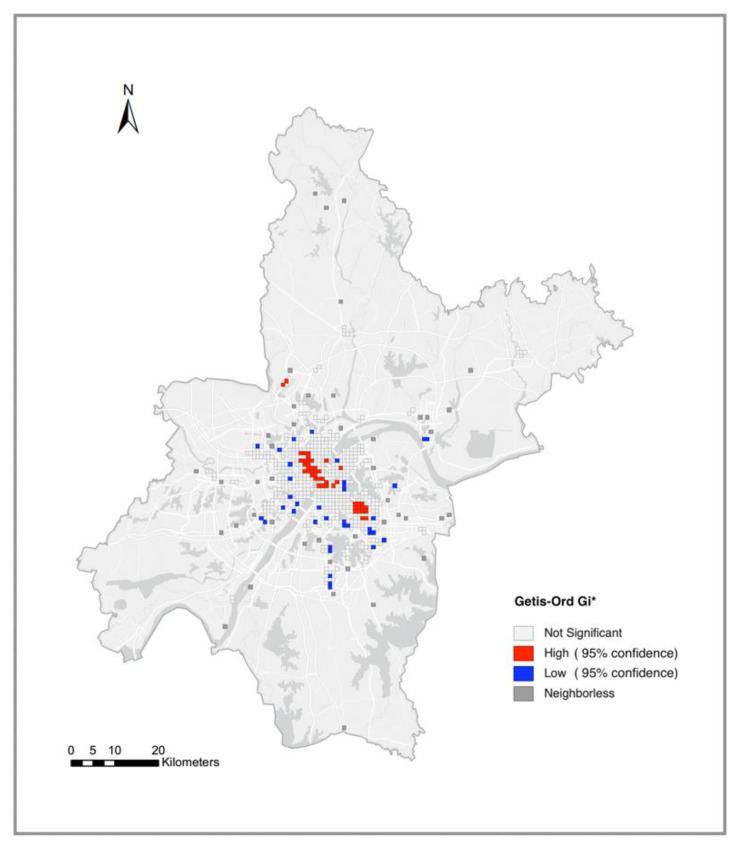
Hot-spot analysis (Getis-Ord Gi*) results of high-risk places in Wuhan.

**Table 1 ijerph-19-06523-t001:** Variable selection.

Theme	Variables	Description
Activities	Proportion of catering (POC)	Proportion of catering activities in total Weibo activity data.
	Proportion of shopping (POS)	Proportion of shopping activities in total Weibo activity data.
	Proportion of recreation (POR)	Proportion of recreational activities in total Weibo activity data.
	Proportion of traffic trips (POT)	Proportion of traffic activities in total Weibo activity data.
	Spatial interaction frequency (SIF)	Spatial interaction frequency between residential space and activity space.
	Duration of outside activities (DOA)	Average outside-activity duration of Weibo users.
	Radius of gyration (ROG)	The radius of gyration measures how far and how frequently Weibo users move.
Demographics	Population density (PD)	The ratio of the resident population to the land area.
	Ageing of population (AOP)	The ratio of population aged over 60 in the total population.

**Table 2 ijerph-19-06523-t002:** Spearman’s correlation results of COVID-19 morbidity rate with the demographic and activity indicators at the county level in Wuhan. PD (population density), AOP (ageing of population), POC (proportion of catering), POS (proportion of shopping), POR (proportion of recreation), POT (proportion of traffic trips), SIF (spatial-interaction frequency), ROG (radius of gyration), and DOA (duration of outside activities).

Variable	Spearman’s *p*	*p*-Value
PD	0.91208	0.000014
AOP	0.73626	0.004107
POC	0.89560	0.000035
POS	0.82967	0.000450
POR	0.34615	0.246625
POT	0.67033	0.012166
SIF	0.80219	0.000968
ROG	0.17033	0.577975
DOA	0.88462	0.000059

**Table 3 ijerph-19-06523-t003:** Kaiser–Meyer–Olkin and Bartlett’s tests.

		Value
KMO	Measure of sampling adequacy	0.810
Bartlett’s Test of Sphericity	Approx. chi-squared	91.951
	Degree of freedom	15
	Significance	0.000

**Table 4 ijerph-19-06523-t004:** Explanatory contribution rates of principal components.

Component	Proportion of Variance	Cumulative Proportion
PC1	81.3271	81.3271
PC2	8.2634	89.5906
PC3	4.1690	93.7595
PC4	3.4327	97.1923
PC5	1.6629	98.8551
PC6	1.0180	99.8731
PC7	0.1260	100.000

**Table 5 ijerph-19-06523-t005:** Component-score coefficient matrix of the first three principal components.

Variable	PC1	PC2	PC3
PD	0.3744	0.4186	0.4253
AOP	0.3511	−0.6389	−0.0319
S I F	0.3596	0.4858	−0.6057
POC	0.3933	0.2405	0.0802
POS	0.3811	−0.1583	0.4068
POT	0.3735	−0.3049	−0.5000
DOA	0.4095	−0.0670	0.1709

**Table 6 ijerph-19-06523-t006:** Summary statistics of OLS and SLM in modeling COVID-19 morbidity rate with the principal components.

	OLS	SLM
Constant	4.6979 *** (0.4019)	1.6195 ** (0.1762)
PC1	0.8059 *** (0.1753)	0.4929 *** (0.1263)
PC2	1.2589 ** (0.5501)	0.9366 *** (0.04067)
PC3	1.2493 (0.7745)	1.0646 ** (0.4752)
w	-	0.6448 *** (0.1762)
R-squared	0.7629	0.8720
Log Likelihood	−20.8809	−17.6773
Akaike info criterion	49.7617	45.3546
Schwarz criterion	52.0215	48.1794
Lagrange Multiplier (lag)	4.6617 **	-
Robust LM (lag)	7.6790 ***	-
Lagrange Multiplier (error)	0.1078	-
Robust LM (error)	3.1251 *	-

Note: * significant at the 0.1 level; ** significant at the 0.05 level; *** significant at the 0.01 level. Standard errors are in parentheses.

**Table 7 ijerph-19-06523-t007:** Regression coefficient of COVID-19 morbidity rate with demographic and activity factors.

Variable	Regression Coefficient
Population density (PD)	0.20136
Ageing of population (AOP)	0.08987
Spatial interaction frequency (SIF)	0.15487
Proportion of catering (POC)	0.17986
Proportion of shopping (POS)	0.15857
Proportion of traffic trips (POT)	0.10396
Duration of outside activities (DOA)	0.16657

## Data Availability

Publicly accessible datasets were analyzed in this study. This data can be found here: https://github.com/yuanmengyue5566/COVID-19-and-Weibo-Data (accessed on 16 April 2020).

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
