# Peer review of "Exploring the Relationship among Human Activities, COVID-19 Morbidity, and At-Risk Areas Using Location-Based Social Media Data: Knowledge about the Early Pandemic Stage in Wuhan"

_ijerph, 2022, doi:10.3390/ijerph19116523_

Round 1
Reviewer 1 Report
- Overall, the paper is well structured and written. The ideas behind the paper are clear. However, the main goal of the study and envisaged results could be fully achieved with a much less complex workflow... Too many redundant and overlapping mathematical models were applied without a real need. This made the workflow more unnecessary and complex than it could be... Mastering and applying lots of complex mathematical methods is nice, but the more important question here is the optimization of the method e.g. the computational time or a reproducibility. For instance, there was no need to use Spearman rank correlation together with OLS regression... It's a bit redundant... There is no need to do both on the dataset... Please optimise your method.
- Further, there is a lack of any information describing the Sina Weibo platform which data was used... Me and probably many of potential readers have no idea what it is... Please introduce it and at least provide a link to it in the footnote.
- In addition, there is no a data specification of Weido data used including data quality details... The only information the authors provided was that according to one of references Sina Weibo provides 'accurate location information'. The readers should know to which extent it is accurate? Dealing with the data without data quality specification provided does not offer credible results in return...
Author Response
Dear reviewer, thank you for the thoughtful review of our work. We summarize major comments you provided and the responses to your points are given below.
1、Overall, the paper is well structured and written. The ideas behind the paper are clear. However, the main goal of the study and envisaged results could be fully achieved with a much less complex workflow... Too many redundant and overlapping mathematical models were applied without a real need. This made the workflow more unnecessary and complex than it could be... Mastering and applying lots of complex mathematical methods is nice, but the more important question here is the optimization of the method e.g. the computational time or a reproducibility. For instance, there was no need to use Spearman rank correlation together with OLS regression... It's a bit redundant... There is no need to do both on the dataset... Please optimise your method.
Response: We agree that using a less complex workflow to obtain the envisaged results would be better. However, using Spearman rank correlation and OLS regression have different emphasis and meaning in our study. Our workflow aims to select relevant variables through Spearman rank correlation and then apply them for subsequent spatial regression modeling. In fact, OLS regression is not the primary method but as a comparison with the spatial regression model to highlight the influence of spatial autocorrelation. This point was probably not presented clearly enough in the previous version of the paper. To clarify this further, we have omitted the method description for OLS in the 2.5 parts (Page 9) and just mentioned it as a comparison with the results of the spatial regression model (Page 9, lines 296-297; Page 14, lines 477-478).
2、Further, there is a lack of any information describing the Sina Weibo platform which data was used... Me and probably many of potential readers have no idea what it is... Please introduce it and at least provide a link to it in the footnote.
Response: Thanks for your viewpoint and we have added 2.3.1 Weibo Data Collecting and Processing part to supplement the introduction of Weibo data (Pages 4-5, lines 135-157).
3、In addition, there is no a data specification of Weido data used including data quality details... The only information the authors provided was that according to one of references Sina Weibo provides 'accurate location information'. The readers should know to which extent it is accurate? Dealing with the data without data quality specification provided does not offer credible results in return…
Response:
Like Twitter, Sina Weibo is also a kind of user generated content, it is really a big challenge for researchers to make data quality assessment of the dataset that applied. Weibo data has shortcomings in data quality such as biased data and sparse distribution. These shortcomings are also introduced and discussed in the conclusion part (page 20, lines 654-661).
Still, check-in data is a high in data quality, most of Weibo check-ins can meet the requirement of four indicators: Readability, Completeness, Usefulness and Trustworthiness. As Fig.3 shows, Weibo check-ins contains user information, text content, time and location (POI) information, which provide essential sptaio-temproal information to analyze human activity patterns in our research. Also, the check-in dataset in our study is pre-processed to avoid noise and invalid records are filtered using the following criteria (page 5, lines 153-157).

Reviewer 2 Report
Dear authors, thank you so much for your nice contribution to the existing body of literature on pandemic tracking with geospatial data sets. I personally liked the content. However, there are some revisions to be resolved before publishing it. I am very much looking forward to reading the next version of this paper. I think, despite the revisions, it is very strong and will be a much needed addition to the literature base.
- "City morbidity" is a vague term. Please use "morbidity patterns in ..." throughout the text.
- Results and findings have to be presented more in the abstract.
- You need to express what your research adds to the existing body of literature and how it confirms or contradicts the previous studies. This should be added at the end of the introduction part in order to make the readers understand the novelty of your research. There have been a vast number of studies that use social media data sets to track the pandemic or investigate the relationship with demographic activities. Even though you discussed your findings well with a comparison with previous studies, this novelty part is a must, I think.
- In the literature review, you need to mention the pandemic isolation tracking projects using geospatial services (as an example of survelliance). Please refer to:
- İban, M. C. (2020). Geospatial Data Science Response to COVID-19 Crisis and Pandemic Isolation Tracking . Turkish Journal of Geosciences , 1 (1) , 1-7 . Retrieved from https://dergipark.org.tr/en/pub/turkgeo/issue/54166/718270
- B. Benreguia, H. Moumen and M. A. Merzoug, "Tracking COVID-19 by Tracking Infectious Trajectories," in IEEE Access, vol. 8, pp. 145242-145255, 2020, doi: 10.1109/ACCESS.2020.3015002.
- Kang, J., Jang, Y. Y., Kim, J., Han, S. H., Lee, K. R., Kim, M., & Eom, J. S. (2020). South Korea's responses to stop the COVID-19 pandemic. American Journal of Infection Control, 48(9), 1080-1086.
- Please use 11.2 million instead of 11212 thousand for the population.
- Please use SI units (km) for the scale bars on all maps rather than using miles.
- Data collection: Since Weibo is not used globally, the readers might need to understand the background of data mining on Weibo. You need to express how you exported geo-coded posts. An API? Or other methods? What was the file format of exported raw data? Tabular, or other formats? How did you clean and filter the raw data? Did you use any programming language to export and clean the raw data?
- Home location: The authors mentioned the users' "home location". How can an analyst know the home location info of any Weibo user? Isn't it private? Or do social media users compulsorily deliver their home location data to social media services in China? Since I am a non-Chinese scholar and I haven't used Weibo, I found this practice very opposite to social media privacy.
- Please add quantitive results in the conclusion part.
- There are several misuses of the English language. In particular, relative clauses and the selection of prepositions are problematic. Sometimes some similar words are used simultaneously, although they have different meanings. A proofread is highly recommended.
- The article numbers in Elsevier journals are formatted as page numbers. Please remove "p." before article numbers in your reference list.
- Please mention the data source of morbidity in administative districts by citing any reference or giving information in the acknowledgements.
Author Response
Thank you for your careful review of our work. Your detailed comments have helped us carry out a major revision of the paper. We have revised the manuscript, and the changes to the manuscript are shown in red. Our responses are given in a point-by-point manner below.
1、"City morbidity" is a vague term. Please use "morbidity patterns in ..." throughout the text.
Response: We agree this term is vague. Thank you for pointing this out and we have changed the term "city morbidity" to "morbidity pattern" in the text (page 1, line 14).
2、Results and findings have to be presented more in the abstract.
Response: We have added more description about the results and findings in the abstract (page 1, lines 21-26): “The results provide statistical evidence regarding the utility of the human activity and demo-graphic factors for the COVID-19 morbidity patterns at the early pandemic stage in Wuhan. The spatial interaction reveals a general transmission pattern in Wuhan and determines the high-risk areas of COVID-19 transmission. This article explores the human activity characteristics from social media check-in data and studies how human activities play a role in COVID-19 transmission in Wuhan.”.
3、You need to express what your research adds to the existing body of literature and how it confirms or contradicts the previous studies. This should be added at the end of the introduction part in order to make the readers understand the novelty of your research. There have been a vast number of studies that use social media data sets to track the pandemic or investigate the relationship with demographic activities. Even though you discussed your findings well with a comparison with previous studies, this novelty part is a must, I think.
Response: Thanks for your valuable comment reminding us to identify our contribution more clearly. We have added the novelty part to elucidate our contribution to the existing literature at the end of the introduction section. (page 2, lines 79-87).
4、In the literature review, you need to mention the pandemic isolation tracking projects using geospatial services (as an example of survelliance).
Response: Thanks for the suggestion on the literature review. We have supplemented the literature on the public health surveillance using geospatial services in the literature review section (page 2, lines 54-55).
5、Please use 11.2 million instead of 11212 thousand for the population.
Response: We have revised "11212 thousand" to "11.2 million" in the text. (page 2, line 92).
6、Please use SI units (km) for the scale bars on all maps rather than using miles.
Response: We have modified the scale bars on all maps to SI units (km) in the revised manuscript.
7、Data collection: Since Weibo is not used globally, the readers might need to understand the background of data mining on Weibo. You need to express how you exported geo-coded posts. An API? Or other methods? What was the file format of exported raw data? Tabular, or other formats? How did you clean and filter the raw data? Did you use any programming language to export and clean the raw data?
Response: Thanks for your viewpoint and we have added 2.3.1 Weibo Data Collecting and Processing part to supplement the introduction of Weibo data (Pages 4-5, lines 135-157). We have added more information about the collection and preprocessing of social media check-in data.
8、Home location: The authors mentioned the users' "home location". How can an analyst know the home location info of any Weibo user? Isn't it private? Or do social media users compulsorily deliver their home location data to social media services in China? Since I am a non-Chinese scholar and I haven't used Weibo, I found this practice very opposite to social media privacy.
Response: It should be noted that the residential information of Weibo users is at the community level and does not involve the user’s detailed home address or house number. We have mentioned this more clearly and reworded some of the descriptions to emphasize this point in the 2.3.3 part (pages 5-6, lines 191-193).
9、Please add quantitive results in the conclusion part.
Response: We’ve added more quantitive results in the conclusion part (page 20, lines 631-642): (i) The human activity indicators charatisticed from Weibo check-in data show statistically significant and positive impacts on the COVID-19 morbidity in Wuhan. The results provide statistical evidence regarding the utility of the human activity indicators (POC, POS, POT, SIF, and DOA) and demographic factors (PD, AOP) for the COVID-19 morbidity patters at the early pandemic stage in Wuhan. (ii) The COVID-19 morbidity pattern at district-level in Wuhan has significant spatial au-tocorrelation. The SLM explains the spatial depandency and obtains a more robust es-timation of the influences of population activity and demographic factors on the COVID-19 morbidity. (iii) The spatial interaction matrix reveals a general transmission pattern within Wuhan and determines the risky areas of COVID-19 transmission.
10、There are several misuses of the English language. In particular, relative clauses and the selection of prepositions are problematic. Sometimes some similar words are used simultaneously, although they have different meanings. A proofread is highly recommended.
Response: We have done our best to correct the misuse of English and some similar words in the revised manuscript. Here is just a simple example, "the risk of COVID-19 infection in each administrative district" was changed to " the morbidity patterns of COVID-19 at the district level " in the text.
11、The article numbers in Elsevier journals are formatted as page numbers. Please remove "p." before article numbers in your reference list.
Response: Thanks for this observation, we have re-reviewed the manuscript and corrected the formatting of all references.
12、Please mention the data source of morbidity in administative districts by citing any reference or giving information in the acknowledgements.
Response: We have added to the acknowledgments mentioning the data source of COVID-19 morbidity in Wuhan (page 20, lines 678-680).

Reviewer 3 Report
The subject of the paper “Exploring the relationship between Human Activities and COVID-19 Morbidity and Risky Areas from Location-Based Social Media Data: Knowledges from the Early Pandemic Time in Wuhan” is timely and valuable to the audience of the IJERPH. Researchers presented results from Weibo users’ check-in trajectories and showed the impact of human mobility on the spread of COVID-19 from the perspective of spatial interaction.
Overall, the paper is well structured, reads quite well, and covers the existing literature quite well. The analysis of the data is interesting and well documented.
To be honest, this is one of the best written and logical structured studies I have read this year. There is everything clear and sound in this paper.
However, one thing that I’m curious about since for the Central-Western Europe reviewer, the issue of personal data privacy is not explained in the paper at all. What are the rules or circumstances under which authors could collect and process data from different individuals on Weibo? How did the authors collect this data? Does Weibo share it publicly? These are important questions since there is nothing about ethics or any data statements in the paper.
I found one obvious mistake, on page 19 supposed to be “source” not “srouce.
Author Response
Thanks for your thoughtful review and kind comments of our work. We summarize major comments you provided and the responses to your points are given below.
1、However, one thing that I’m curious about since for the Central-Western Europe reviewer, the issue of personal data privacy is not explained in the paper at all. What are the rules or circumstances under which authors could collect and process data from different individuals on Weibo? How did the authors collect this data? Does Weibo share it publicly? These are important questions since there is nothing about ethics or any data statements in the paper.
Response: Thanks for the valuable comments, and we have added the 2.3.1 Weibo Data Collecting and Processing part to supplement the introduction of Weibo data (Pages 4-5, lines 135-157). We have added more information about social media check-in data collection and preprocessing. Note that the data in our research was exported from a public source. We collected Weibo check-in data via Sina Weibo’s open APIs (http://open.weibo.com). The issue of data privacy has been stated in the 2.3.1 part (page 5, lines 145-146).
2、I found one obvious mistake, on page 19 supposed to be “source” not “srouce”.
Response: Thank you for pointing this out and we have corrected the word "source". (page 19, line 628).

Round 2
Reviewer 1 Report
The manuscript has been sufficiently improved according to my previous review report.